# Breast Cancer and Anaesthesia: Genetic Influence

**DOI:** 10.3390/ijms22147653

**Published:** 2021-07-17

**Authors:** Aida Raigon Ponferrada, Jose Luis Guerrero Orriach, Juan Carlos Molina Ruiz, Salvador Romero Molina, Aurelio Gómez Luque, Jose Cruz Mañas

**Affiliations:** 1Institute of Biomedical Research in Malaga (IBIMA), 29010 Malaga, Spain; aidaraigonp@gmail.com (A.R.P.); jagomez@uma.es (A.G.L.); 2Department of Anaesthesiology, Virgen de la Victoria University Hospital, 29010 Malaga, Spain; jcrm92_8@hotmail.com (J.C.M.R.); salromo16@gmail.com (S.R.M.); jose.cruz.sspa@juntadeandalucia.es (J.C.M.); 3Department of Pharmacology and Pediatrics, School of Medicine, University of Malaga, 29010 Malaga, Spain

**Keywords:** anaesthetic drugs and techniques, opioids, propofol, volatile agent, breast cancer, cancer recurrence, biomarkers, miRNA

## Abstract

Breast cancer is the leading cause of mortality in women. It is a heterogeneous disease with a high degree of inter-subject variability even in patients with the same type of tumor, with individualized medicine having acquired significant relevance in this field. The clinical and morphological heterogeneity of the different types of breast tumors has led to a diversity of staging and classification systems. Thus, these tumors show wide variability in genetic expression and prognostic biomarkers. Surgical treatment is essential in the management of these patients. However, the perioperative period has been found to significantly influence survival and cancer recurrence. There is growing interest in the pro-tumoral effect of different anaesthetic and analgesic agents used intraoperatively and their relationship with metastatic progression. There is cumulative evidence of the influence of anaesthetic techniques on the physiopathological mechanisms of survival and growth of the residual neoplastic cells released during surgery. Prospective randomized clinical trials are needed to obtain quality evidence on the relationship between cancer and anaesthesia. This document summarizes the evidence currently available about the effects of the anaesthetic agents and techniques used in primary cancer surgery and long-term oncologic outcomes, and the biomolecular mechanisms involved in their interaction.

## 1. Introduction

Breast cancer is the leading cause of mortality in women [1]. Even when the tumor is completely resected, tumor recurrence occurs in up to one third of patients, with metastatic disease being the direct cause of death [2]. Surgery may generate systemic inflammatory response syndrome, which causes oxidative stress, and in turn impairs the anti-tumor immunologic response [3]. Surgical stress activates a neuroendocrine response in the hypothalamic–pituitary–adrenal axis (HPA axis) and sympathetic nervous system (SNS), which results in the suppression of cell-mediated immunity (CMI); this suppression is induced by the release of neuroendocrine mediators such as catecholamines, cortisol, and cytokines [4].

These mediators, including vascular endothelial growth factor (VEGF), matrix metalloproteinases (MMPs) and interleukin (IL) 6 and 8, are endogenous regulators that promote tumor growth and angiogenesis, thereby favoring metastasis.

Recent studies reveal that the type of anaesthesia administered during cancer surgery may influence the course of the disease [5]. However, there is limited evidence available that supports the modification of standard anaesthetic approaches.

Further studies are necessary to identify the mechanisms by which each agent interacts with tumor cells. The aim of this article is to provide a review of the current knowledge on how anaesthetic drugs and techniques affect breast cancer recurrence.

## 2. Anaesthetics and Cancer Relapse

### 2.1. Hypnotics

#### Propofol

Propofol may have beneficial effects on survival in cancer patients, including breast cancer patients. This agent inhibits tumor cell migration and proliferation, promotes tumor apoptosis and has anti-inflammatory activity [5,6,7]. This agent acts on the immune system at the level of natural killer lymphocytes (NK), which belong to innate immunity. Cho et al. [8] conducted a prospective study comparing a group of patients who received propofol-ketorolac vs. sevoflurane-fentanyl. The authors observed a reduction in NK activity in the sevoflurane group, whereas this activity was increased in the propofol group. Although evidence consistently shows that propofol has tumor-killing activity, two recent studies associate it with pro-tumor activity in breast and bladder cancer, mediated by the activation of the Nrf2 pathway and the reduction in p53 levels [9,10].

Propofol favors tumor cell apoptosis by affecting matrix metalloproteinase (MMPs) expression, which play a crucial role in extracellular protein degradation and epithelial-mesenchymal transition (EMT), activate vascular endothelial growth factor (VEGF) [11], and inhibit intrinsic apoptosis pathways [12]. Two pathways have been identified to be involved in the inhibition of the synthesis of these proteins: the MAP-kinase pathway (ERK1/2, JNK y p38) in colon cancer [13], and NF-κB in breast cancer [14].

As for the ability to induce tumor apoptosis, several pathways have been investigated, such as the inhibition of anti-apoptosis mechanisms, including Bcl-2, Sox4, Akt/mTOR and Wnt/β-catenin, and the increase in pathways involving tumor suppressor genes such as the Bax, ING3, Fox01 and caspase pathways [15,16,17,18,19,20]. In addition, Wang et al. [21] demonstrated that propofol induces the intrinsic apoptotic signaling pathway by the release of reactive oxygen species (ORS).

Propofol may inhibit surgery-induced systemic inflammatory response syndrome throughout decreasing cell concentrations of hypoxia-inducible factor 1 (HIF1A), which is elevated in the tumor’s hypoxic micro-environment and promotes cell migration and invasion [22]. Ecimovic et al. [23] demonstrated in vitro the role of NET-1 (neuroepithelial cell transforming gene-1) overexpression, a gene that is associated with tumor dissemination and which decreases with propofol exposure [24].

### 2.2. Halogenated

It has been suggested that inhalation agents, for the most part, have a tumorigenic effect since they both inhibit tumor cell apoptosis and stimulate tumor cell proliferation and migration. More specifically, in advanced breast cancer, elevated caveolin-1 concentrations have been associated with lower survival. This protein has been linked to higher resistance to apoptosis, migration and elevated invasivity in breast cancer cells. Ecimovic et al. [25] analyzed, in vitro, the ability of sevoflurane to stimulate tumor cell proliferation, migration and invasion in patients with positive (ER+) and negative (ER-) estrogen receptor breast cancer (with the latter not having invasive capacity).

However, Kawaraguchi et al. [26] documented that isoflurane confers a protective effect on tumor cells in colon cancer against TNF-mediated apoptosis (TRAIL or TNF-related apoptosis-inducing ligand) by interacting with caveolin-1.

Sevoflurane reduces NK activity, thereby reducing immunosurveillance and favoring progression of micrometastases. In contrast, a range of studies have been conducted to compare immunosuppression induced by halogenated anaesthetics vs. general intravenous anaesthesia, with inconsistent results [27,28,29,30,31,32]. Enlund et al. [27] found no significant differences in 1-year and 5-year survival in a sample of 1837 breast cancer patients. Kim et al. [29] compared propofol with a variety of halogenated agents (sevoflurane, desflurane, isoflurane and enflurane), without significant differences. Recent retrospective studies [30,31,32] provide cumulative evidence of the absence of significant differences between intravenous and inhalation anaesthetics in terms of recurrence or survival.

## 3. Analgesics

### 3.1. Opioids

Opioids have an immunosuppressive effect that influences cellular and humoral immunity, as they reduce NK lymphocyte activity and proliferation, citokine production, phagocytic activity, and antibody release [33]. The type and degree of immunosuppression depends on the type, dose and time of exposure to the opioid. All synthetic opioids reduce NK activity [34,35].

Morphine inhibits T and NK lymphocyte activity, promotes lymphocyte apoptosis, reduces toll-like 4 factor in macrophages [36] and has angiogenic activity [37]. In addition, the tumorigenic activity of morphine is mediated by two independent mechanisms, namely: by direct stimulation of mu receptors in tumor cells, the overexpression of which has been associated with poor prognosis, and indirectly by promoting neo-angiogenesis through metabolic signaling pathways similar to those used by VEGF factor [38,39].

In breast tumor cells, fentanyl exhibits an antitumoral effect by reducing levels of proteins involved in cell apoptosis and differentiation mechanisms (Bax, Bcl2, Oct4, Sox2, and Nanog) [40]. Although tramadol is a µ-receptor agonist, its analgesic effect is prevailingly mediated by the inhibition of noradrenaline and serotonin reuptake. Sacerdote et al. [41] assessed the relationship between tramadol and immune response in patients with uterine carcinoma. The authors found that tramadol not only inhibits but also stimulates NK lymphocyte activity. Thereupon, Xia et al. [42] demonstrated, in vitro, in breast tumor cells that tramadol reduces tumor cell proliferation, migration and invasion by up to 28 days through the inhibition of the α2-adrenergic receptor. In a retrospective study, Kim et al. [43] observed lower rates of mortality and tumor recurrence in the group of breast cancer patients treated surgically who received tramadol. This effect is conferred by the inhibition of tumor cell proliferation, induction of apoptosis, and action on serotonergic receptors and transient receptor potential channel V1 or TRPV1.

### 3.2. Regional Anaesthesia and Local Anaesthetics

There is a variety of locoregional anaesthesia techniques in breast cancer surgery that have good analgesic outcomes. Paravertebral block (PVB) is the most widely used technique, although it is associated with a higher risk of severe complications. New techniques have been developed, with pectoral block type II having shown good effectiveness, and having been employed in a similar context as PVB [44]. Several studies have been published comparing general anaesthesia and combined anaesthesia: five retrospective studies, two prospective studies and a systematic review. These studies provide evidence of the beneficial effects of not using opioids and/or local anaesthetics per se [45,46,47,48,49,50,51,52].

Exadaktylos et al. [45] published the first retrospective study assessing the outcomes of 129 patients with breast cancer treated surgically, of whom 50 received combined anaesthesia (PVB+Propofol) and 79 balanced general anaesthesia. The rate of recurrence was lower in the group that received combined anaesthesia. In contrast, a recent systematic review conducted by Pérez-González et al. [52] did not show statistically significant differences between combined and general anaesthesia.

Local anaesthetics block afferent and efferent nerve response and effectively suppress sympathic stimulation through the inhibition of hypothalamic–pituitary–adrenal (HPA) stimulation induced by surgical stress, thereby reducing HPA activity [52].

Lidocaine has been proven to exert beneficial effects in vivo and in vitro, and it is associated with a reduction in tumor cell proliferation, migration and invasion in breast, liver and lung cancer [53,54,55]. Lidocaine inhibits the proto-oncogen that releases Src, an intracellular non- tyrosine kinase protein that is involved in cell proliferation and migration processes through ICAM-1 phosphorylation, which enables neutrophils to cross the endothelium and increase immune response [55]. It has been reported to have effects on other signaling pathways such as TRPV-6 inhibition [56] or DNA demethylation in breast cancer cells [57]. Chang et al. [58] demonstrated in vitro that both lidocaine and bupivacaine induce breast cancer cell apopotosis through the activation of caspases 7, 8 and 9. In the same line, D’Agositino et al. showed that lidocaine inhibits cytoskeletal modification in breast cancer cells [59].

Evidence has been provided that lidocaine infiltration in the peritumoral region inhibits tumor growth by binding EGFR [60]. As for the immune system, lidocaine, at clinically relevant concentrations, stimulates the cytotoxic effect of NK lymphocytes [61]. A prospective, randomized trial conducted by Galoș et al. revealed that lidocaine reduced neutrophil extracellular traps, a phenomenon that has been associated with tumor recurrence [62].

### 3.3. NSAIDs

The enzyme cyclooxygenase (COX-2) causes an increase in prostaglandins, which are involved in immune system control and angiogenesis. Ketorolac is the most extensively studied NSAID in relation to cancer. It is a COX-1 and COX-2 inhibitor that is commonly used in the perioperative period. Evidence from retrospective studies demonstrates that perioperative administration of ketorolac reduces breast cancer recurrence by diminishing the production of prostaglandins and VEGF. Forget et al. attempted to replicate these results in patients with breast cancer at high risk of recurrence (triple negative, neutrophil/lymphocyte ratio ≥ 4) [63] in a prospective study [64] of 203 patients, without differences having been found between treatment groups.

## 4. Dexmedetomidine

Dexmedetomidine is a selective α2 agonist with sympatholytic and anti-inflammatory activity that reduces IL-6, IL-8 and TNF-α concentrations and increases anti-inflammatory cytokine IL-10 levels [65].

Despite its anti-inflammatory effect, a pro-tumoral activity is attributed to dexmedetomidine. Lavon et al. [66] demonstrated in animal models that it promotes metastasis in breast, lung and colon cancer. This effect is credited to the transient immunosuppression induced by dexmedetomidine, added to the effects of surgical stress and changes in vascular patency.

In the same line, Xia et al. [67] investigated the effect of dexmedetomidine in breast cancer cells, in vitro and in vivo, in mice and concluded that dexmedetomidine promotes tumor cell proliferation, migration and invasion through the inhibition of the α2/ERK adrenergic receptor pathway [42].

Cata et al. [68] performed a retrospective study involving 1404 patients with non-small lung cancer (NSCLC) treated surgically to investigate a potential relationship between tumor recurrence and the use of dexmedetomidine. This relationship was not confirmed. Indeed, the results showed a significant relationship between dexmedetomidine and lower survival.

Beta-blockers and lipid lowering drugs are two of the main groups of drugs among patients undergoing a surgical procedure.

## 5. Beta-Blockers

Beta-adrenergic receptors are found both in tumor cells and the immune system [69], and seem to play a key role in carcinogenesis [70]. Beta-blockers have been proven to be involved in angiogenesis and cellular neoproliferation [71]. Exposure to beta-agonists inhibits lymphocyte NK activity [72,73] and induces an increase in T-regulator lymphocytes [74], leading to immunosuppression. Kang et al. documented that adrenergic stimulation activated the MAP-kinase cascade and, more specifically, the DUSP1 cascade, which causes resistance to chemotherapy and apoptosis [75]. Recently, Zhou et al. [76] observed that propranolol prevented T-regulator lymphocyte elevation. Another potential cellular signaling pathway is adrenergic activation of PI3K/AKT and HIF-1 α, which is also inhibited by propranolol [77].

Contradictory results were obtained in five retrospective [78,79,80,81,82] and two cohort studies [83,84] assessing recurrence in breast cancer patients after surgery due to the lack of a standard treatment administration protocol [85,86,87,88].

## 6. Lipid Lowering Drugs

The increased prevalence of cardiovascular disease in the recent years has resulted in an increase in the use of lipid lowering drugs, with statins being the most common pharmaceutical group. As a component of the cellular membrane, cholesterol plays an essential role in cellular division; therefore, a reduction in extracellular cholesterol should cause an inhibition of tumor cell proliferation. Cholesterol metabolites such as 27-hydroxycholesterol and 25-hydroxycholesterol may stimulate estrogen receptors (ERs) [89,90]. Alikhani et al. [91] reported an increase in breast tumor growth mediated by the PI3K/AKT pathway in hyperlipidemic mice.

Cholesterol favors a pro-inflammatory environment by the activation of macrophage toll-like receptors [92] and the inhibition of CCR7 expression in dendritic cells, which explains their antigenic effects [93]. On the other hand, cholesterol modulates lymphocyte T activity through the liver X receptor (LXR) [94].

As for the use of statins, inconsistent results were obtained in six retrospective [80,95,96,97,98,99] and five prospective studies [85,100,101,102,103] (four supporting its use and the remaining seven having not provided clinically relevant results). In contrast, the three meta-analyses retrieved [103,104,105,106] provide consistent evidence that statins reduce breast cancer recurrence. However, these studies were conducted using non-standard methods, and prospective randomized studies are needed.

## 7. Biomarkers, Anaesthetic Technique and Cancer

In previous paragraphs, we have described different routes and mechanisms by which drugs administered in the perioperative period affect breast cancer recurrence. In the next section, we will discuss the current knowledge of how these drugs and techniques affect the most important biomarkers, as well as their clinical relevance.

Early detection, disease monitoring during treatment and subsequent follow-up to detect recurrences are still critical for improving the outcomes, morbidity and mortality of cancer patients. This is hampered by the absence of obvious symptoms and insufficiently sensitive biomarkers. Biopsy and imaging examination applications are limited; in addition, traditional tumor diagnostic markers exhibit low sensitivity. Therefore, there is an exponential development of evidence for the detection and validation of novel, more sensitive, and easy-to-detect biomarkers which can be used in the diagnosis and prognosis of cancers.

A multiplicity of anaesthetic techniques are used in clinical practice in patients undergoing cancer surgery. The pharmacokinetics and pharmacodynamics of common anaesthetic agents are thoroughly understood and their use is determined based on the characteristics of each patient. On the other hand, the retrospective analysis of the results in oncological surgery showed that the use of total intravenous anaesthesia (TIVA) was associated with improved recurrence-free survival and overall survival [107].

Thanks to the growing evidence, special attention has been given to the molecules having remarkable ability to target oncogenic protein network, restore drug sensitivity and induce apoptosis in cancer cells. The mechanisms through which general anaesthetics modulated wide-ranging deregulated cell signaling pathways involved in angiogenesis, metastasis, drug resistance, immune-related responses, cytokine secretion, proliferation, invasion and prognosis remained unclear.

The beneficial effects and clinical implications of the different pharmacological groups in cardiovascular surgery are widely known. These include the direct organ-protective effects of sevoflurane mediated by the modulation of metabolic cascades [108,109,110]. In neuroscience, the analysis of the mechanisms of effects of isoflurane on the brain for short periods showed that differentially expressed genes, such as homeobox containing 1, APC, MAPK1 and miRNA-17-5p, were involved [111].

Propofol is one of the most studied drugs in onco-anaesthesia, and there is evidence of its ability to modulate molecular targets such as the Wnt/β-catenin, JAK/STAT and mTOR-driven pathways [112]. Targeted therapies based on molecular markers have potential for the diagnosis and prognosis of a diversity of diseases, including cancer. The development of drugs that act on these molecular pathways show promising results in breast cancer [113].

The identification of biomarkers helps to better understand the influence of the anaesthetic technique on tumor progression and select the optimal anaesthetic plan for each cancer patient. For example, exposure to halogenated agents increased Lipocalin 2 expression over a prolonged period of time [114], and this over-expression also promotes cell migration and invasion through activating ERK signaling to increase SLUG [115], whereas knockdown suppresses the growth and invasion of cancer cells [116]. On the contrary, propofol inhibits growth and invasion through the regulation of the Slug signaling pathway [117].

Lipocalin 2 (LCN2), also known as neutrophil gelatinase associated lipocalin (NGAL), regulates angiogenesis [118,119] and has been reported to be elevated in several types of cancer, including breast cancer. The evidence obtained in vitro and in vivo demonstrates that the tumorigenic and metastatic potential of Lcn2 is induced by TEM promotion, cell migration, invasion, VEGF production, and angiogenesis. The mechanisms that mediate the metastatic and oncogenic potential of Lcn2 include the PI3K/Akt/NF-κB and HIF-1a/ERK signaling pathways, and the protective formation of the MMP-9/Lcn2 complex [120,121].

When its activity is knocked down, angiogenesis decreases through the inhibition of the MMP-9/Lcn complex, thereby reducing MMP-9 activity, cell migration and invasion capacity [118,119,121,122,123,124,125]. In addition, ICAM-targeted Lcn2 sRNA (small interfering RNA) significantly reduced the angiogenic activity and migration of triple-negative breast tumor cells [126,127]. Leng et al. reported the use of anti-Lcn2I policlonal antibodies and they achieved the inhibition of 4T1 cellular metastases [128].

A further alternative molecule that has been associated with the interaction between anaesthetics and cancer is the endothelial adhesion molecule. The main function of endothelial adhesion molecules is mediated by immune response. A range of studies demonstrate that intercellular adhesion molecule-1 (ICAM-1) triggers the activation of multiple cellular signaling pathways that promote tumor cell proliferation, migration and resistance to apoptosis, as well as the development of drug resistance induced by cellular adhesion molecules [129,130]. Where inflammation occurs, the activated leukocytes adhere to the endothelium through ICAM-1, and cell transmigration follows through vessel walls. A similar mechanism mediates extravasation of circulating tumor cells during metastasis, where endothelial ICAM-1 might play a crucial role [129].

ICAM-1 is expressed in several types of tumors and, through the mediation of its main ligand, LFA-1, it may facilitate immunosurveillance [131,132]. Several studies associate ICAM-1 expression with a more aggressive tumor phenotype and higher metastatic potential [133,134]. The capacity of infiltration of breast cancer cells is related to ICAM-1 expression [130]. Serum levels are elevated in breast cancer patients [135] and correlate with advanced-stage or recurrent disease [136,137] and metastasis. Schröder et al. documented the association between ICAM-1 mRNA in breast cancer, high levels of urokine plasminogen (uPA) and uPA-1 inhibitor (PAI 1), and Ki67 and VEGF mRNA overexpression [134].

Expression of ICAM-1 is upregulated by tumor necrosis factor-alfa (TNF-α); meanwhile, local anaesthetics may have an anti-inflammatory effect, since they effectively block the inflammatory signaling of TNFα. The resulting decrease in Akt, endothelial nitric oxide synthase, and Src phosphorylation reduced neutrophil adhesion and endothelial hyperpermeability. Lidocaine and ropivacaine also impair cancer cell migration by inhibiting Src activation that is induced by TNF-α and the phosphorylation of ICAM-1 [138,139].

The roles of the immune system in cancer, from tumor initiation to metastatic progression, remain unclear. Neutrophil extracellular traps (NETosis) form part of the immune response to cancer antigens. NETosis originates from neutrophil degranulation, which results in the release of neutrophil contents into the bloodstream, where they can be detected [140,141]. NETosis processes have been described in the tumor microenvironment [142,143].

Tumor cells can interact direct or indirectly with neutrophils to induce NET production. Intra-and inter-cellular molecular communications generate micro-environments that act as a premetastatic niche to recruit primary tumor cells and form multiple metastases [144,145]. Park et al., described the relationship between metastatic cancer and NET formation through the use of anti-G-CSF blocking antibodies, which reduce the ability of 4T1 cells to induce NETosis [146].

The close cooperation between tumor cells, neutrophils and NETosis in the tumor microenvironment demonstrates the role of NETosis in cancer progression and metastasis [143,147,148]. The myeloperoxidase enzyme (MPO) and citrullinated histone-3 are specific proteins released during NETosis and can be easily measured in serum [143]. The release of chromatin by tumor cells and neutrophils influences tumor progression and thrombosis associated with cancer [149].

This phenomenon is associated with poor prognosis in esophageal cancer and a higher risk for tumor infiltration in patients with higher NETosis activity [150]. The presence of circulating NETs has been described, in patients with advanced esophageal, gastric and lung cancer, at higher levels than healthy controls. The authors further demonstrated, in murine models, that NETs can regulate disease progression, since blocking their formation reduced the occurrence of metastases [151].

Firstly, the relationship between NETs and anaesthetics emerges in patients with septic shock. Propofol shows that clinically relevant concentrations inhibit ROS-dependent NET production by human neutrophils [152]. Chen et al., compare the effects of four intravenous anaesthetics. Through this study, they define a new anti-inflammatory effect of intravenous anaesthetics. Propofol appears to decrease NET formation by activated neutrophils through the inhibition of hypochlorus acid production and MAPK/ERK [153]. Local anaesthetics in clinically used doses, decreased, in a dose-dependent manner, the time to onset of maximal ROS production and half-maximal NETosis [154].

Two randomized clinical trials have been conducted to assess the influence of different anaesthetic techniques in NETosis. Aghamelu et al. found no significant differences between the halogenated agents and propofol in patients with breast cancer treated surgically [155] In addition, Galos et al. incorporated the differential use of intravenous lidocaine in their treatment protocol and observed significant differences in NETosis markers between groups [62].

### 7.1. miRNAs and Drugs Used during Anaesthetic Procedures

Even though enzymes play a key role in describing the effects of anaesthetic drugs and techniques on breast cancer recurrence, the codifying process may also have major importance in this issue. Non-coding RNAs, particularly MicroRNAs, are playing a growing role in the field of oncology and anaesthesia.

There is growing interest in the identification of biomarkers for assessing individual risk of cancer progression and response to treatment. MicroRNAs (miRNAs) are endogenous non-coding RNA sequences that regulate gene expression and suppress or activate multiple genes at pre- and post-transcriptional levels [156,157]. In recent years, evidence of the relevant role of miRNA in the molecular mechanisms of carcinogenesis in healthy patients has accumulated [158,159,160,161]. Although these changes are detectable in tumor tissue, levels of deregulated miRNAs can also be determined in blood [162]. Likewise, there is wide inter-subject variability in deregulated miRNA levels in healthy subjects and in cancer patients treated surgically [163].

Current evidence suggests that the beneficial or deleterious effects of anaesthetics during cancer surgery are mediated by genetic and molecular mechanisms [164]. Multiple studies have been published on the interaction of propofol with the genetic mechanisms that mediate miRNA suppression/overexpression in different types of tumors: propofol increases miRNA let7 expression and apoptosis in ovarian cancer cells [165]; it increases miRNA 218 and miRNA 451 expression, thereby resulting in the reduction in MMP-2 expression and cellular proliferation [166]. Finally, propofol reduces the invasive capacity of liver cell carcinoma through miRNA-199a and its action on MMP-9 [167].

Sun et al. concluded that propofol reduces miRNA-374a expression and modulates the forkhead box O1 pathway response to reduce proliferation and resistance to cisplatin in ovarian cancer cells [168].

Propofol has demonstrated, in vitro and in vivo, its ability to facilitate sensitivization to trastuzumab-resistant tumor cells through epigenetic changes mediated by the miRNA-149-5p/IL-6 axis [169]. Propofol induces apoptosis in breast cancer cell lines through the inactivation of the miRNA-24/p27 signaling pathway [170]. Propofol has also been associated with one of the major types of onco-miRNA-2; it is overexpressed in early-stage pancreatic cancer. The action of propofol on miRNA-21 involves the suppression of Slugh expression, thereby increasing the proapoptotic effect of the p53 gene and inducing e-cadherin overexpression [171].

Buschmann et al. recently published the first prospective study comparing the serum of patients with colon cancer—who were treated surgically and exposed to propofol or sevoflurane—and miRNA expression patterns. Significant differences were observed between groups, with propofol showing an inhibitory effect on signaling pathways involved in carcinogenesis [172]. Deng et al. exposed breast cancer cell lines to sevofluorane vs. propofol, and observed an increase in tumor survival in the propofol group, mediated by the modulation of intracellular calcium homeostasis [173]. In contrast, Liu et al. observed that sevoflurane suppressed breast cancer cell proliferation through an increase in miRNA-203 overexpression [174]. Its potential anti-tumor activity is demonstrated by miRNA-155 suppression in papillary thyroid carcinoma [175].

Interactions have been observed in other drugs commonly used in clinical practice. Li et al. documented that ropivacaine inhibits cellular viability and induces apoptosis, whereas concomitant use of sufentanil mitigates its action by means of miRNA-182-5p/BCL10/CYCS regulation [176]. Qu et al. suggest that the beneficial effects of lidocaine on colorectal cancer could be mediated by the upregulation of miRNA-520 a-3p and targeting EGFR [177].

Sequencing of the genome of macrophages treated with morphine shows a differential expression pattern of 26 miRNAs. Of these, miRNA-15b-5p and miRNA-181b-5p are involved in inflammatory and infectious processes, thereby promoting the suppression of immune functions [178]. A diversity of studies reveals the upregulation of 9 miRNAs and downregulation of 17 miRNAs. More specifically, the miRNAs of the Let-7 family have been found to be associated with the MOR pathway [179,180,181]. Low doses of morphine promote non-differentiated cell proliferation, thus reducing miRNA-133b expression. In contrast, at high doses, they have the opposite effect and reduce miRNA-133b and miRNA-128 levels; this effect cannot be countered by naloxone, which suggests an independent mediation by MOR [182]. Exposure to morphine upregulates miRNA-543 expression and contributes to tumor progression through MARCKs inhibition [183].

#### 7.1.1. MiRNAs and Breast Cancer

Different studies document the association between breast cancer and a variety of miRNA expression patterns, which enables the analysis, classification and characterization of different types of tumors [184,185,186]. Mar-Aguilar et al. reported the potential usefulness of three miARNs in breast cancer screening [187]. The anti-tumor effect of propofol and pro-tumor effect of sevoflurane have been demonstrated in in vitro studies. These effects are mediated by the modulation of miRNA- 24/p27 and miRNA-203 expression, respectively [170,174].

#### 7.1.2. MiRNA-21

There is consistent evidence supporting the oncogenic role of miR-21, which is involved in the inhibition of tumor cell apoptosis and contributes to tumor growth. MiR-21 is significantly overexpressed in a wide variety of solid tumors, including breast cancer. MiR-21 overexpression has been associated with tumor progression and poor prognosis [188]. It is involved in the inhibition of tropomyosin alpha-1, the TP53 and TAp63 pathways (regulating LKB1 transcription) [189], programmed cell death 4 (related to apoptosis) [190], and phosphatase tensin homolog (PTEN) [191,192,193,194].

In vitro studies reveal that miR-21 is overexpressed in tissue and cells of triple-negative breast cancer models. This confirms its relationship with MDA-MB-468 cell proliferation and invasion, and the regulation of PTEN expression [195]. MiR-21 knockdown reduced proliferation, viability and invasivity, demonstrating that PTEN is a target of this mRNA [196]. Its overexpression is also linked to poor survival due to the development of resistance to trastuzumab [197,198,199].

#### 7.1.3. MicroRNA-202

MiR-202 has been recently associated with multiple types of cancer, including breast cancer. It belongs to the Let-7 miRNA family, and its downregulation has been associated with poor differentiation and higher tumor aggressiveness [200].

The let-7 family regulates estrogen receptor alpha in breast cancer growth [201] and is involved in the self-renovation and tumorigenicity of breast cancer cells [202]. Joosse et al. detected a relationship between miR-202 levels and prognosis, which suggests a potential role in tumor staging [203].

#### 7.1.4. MicroRNA-155

MiRNA-155 is involved in immune function through cancer development and the production of cytokines. It is overexpressed in different types of cancer, including breast, liver, uterine and lung cancer [204,205,206]. It favors breast cancer progression by acting on TP53 [207,208] and accelerating metastasis through tumor protein p53-inducible nuclear protein 1 (TP53INP1) [209].

## 8. Conclusions

The perioperative use of anaesthetic/analgesic techniques with protective effects on anti- metastatic immune response may reduce tumor progression. A limitation of most studies published to date is that they were performed in vitro, where conditions are not the same as in vivo. The use of biomarkers could be useful to identify scenarios in which there is a high risk of cancer progression. Randomized clinical trials are required to investigate whether anesthetic/analgesic techniques and anaesthetic agents are effective in reducing cancer relapse and increasing survival in oncological patients.

## Data Availability

Not applicable.

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
