# Peer review of "Breast Cancer and Anaesthesia: Genetic Influence"

_ijms, 2021, doi:10.3390/ijms22147653_

Round 1
Reviewer 1 Report
The manuscript, entitled “Breast cancer and anaesthesia: genetic influence”, summarizes the evidence currently available about the effects of the anesthetic agents and techniques used in primary cancer surgery and long-term oncologic outcomes, and the biomolecular mechanisms involved in their interaction. In this review, the authors enumerated proofs of the influence of anesthetic techniques on the physiopathological mechanisms of survival and growth of the residual neoplastic cells released during surgery. This review has potential guiding significance for the clinical treatment of cancer. However, specific issues suggested for attention include:
Major issues:
- In this study, the aim is to review the connection between breast cancer and anesthesia. However, in “6. Biomarkers, anesthetic technique and cancer.”, what is the relationship between these molecules and anesthesia in cancer patients? The author should describe more details in this part.
- Line 86-88, what is the result/conclusion.
Minor issue:
- Correct English spelling and grammar. Such as in abstract, line 11, “relevant” change into “relevance”.
- Some sentences are not well organized.
1) Line 71-74 lead to misunderstanding, should organize again.
2) The paragraphs in Line 86-90 should move to the end of the last paragraph (Line 85). 3) Line 118-119 lead to misunderstanding.
4) The paragraphs in Line 106-114 should move to the end of the last paragraph (Line 105).
- Line 71, “SIRS” in the full name.
Reviewer 2 Report
The work entitled “Breast cancer and anesthesia: genetic influence” describe the current literature about the effect of anesthesia in breast cancer.
A lot of literature were revised in this manuscript; however, the reading has no flow. It seems that the sentences of each author reference were just put it together but no connectivity, starting in the abstract, continuing throughout the manuscript.
In the introduction I would change the first reference for Siegel, 2021 Cancer Statistics 2021.
Some grammar and format have to be verified. A lot of spaces between last word and comma (examples: lines 103, 107, 147, 150 and so on).
Review references. The references in the text are not matching with the reference list, starting reference 41 (could be before, but 41 has the name).
Round 2
Reviewer 2 Report
Line 267 is missing a 'the' and a comma. "...Now, in the next section, we will discuss how these drugs (not this drugs).
Line 268 should be and their clinical relevance instead of its (you need to use plural)
Line 271 "... improving outcomes, morbidity and mortality..."
Line 283 - define what is TIVA
Line 296 - there is no reason to add a dot after express genes. "... expressed genes, such as homeobox..." and remove were involved.
Line 300 - Propofol is one of the most studied drug in...evidence related to its ability to modulate"
Line 310 - missing a comma "... SLUG, whereas.."
Line 351 - missing a comma "... ICAM-1, meanwhile.." Also, this phrase should be rewritten, there is no connection with the first sentence with the rest.
Line 404 - the section 6 is duplicated
Line 582 - "Propofol has also been associated with one of the major onco-miRNA-2, which is overexpressed in early stage..."
